# Electrostatic potentials of atomic nanostructures at metal surfaces quantified by scanning quantum dot microscopy

Rustem Bolat[1,2,3], Jose M. Guevara[1], Philipp Leinen[1], Marvin Knol [1,2,3], Hadi H. Arefi [1,2], Michael Maiworm[4], Rolf Findeisen[4], Ruslan Temirov [1,2,5], Oliver T. Hofmann[6], Reinhard J. Maurer [7,8], F. Stefan Tautz [1,2,3] & Christian Wagner [1,2] ✉

The discrete and charge-separated nature of matter — electrons and nuclei — results in local electrostatic fields that are ubiquitous in nanoscale structures and relevant in catalysis, nanoelectronics and quantum nanoscience. Surface-averaging techniques provide only limited experimental access to these potentials, which are determined by the shape, material, and environment of the nanostructure. Here, we image the potential over adatoms, chains, and clusters of Ag and Au atoms assembled on Ag(111) and quantify their surface dipole moments. By focusing on the total charge density, these data establish a benchmark for theory. Our density functional theory calculations show a very good agreement with experiment and allow a deeper analysis of the dipole formation mechanisms, their dependence on fundamental atomic properties and on the shape of the nanostructures. We formulate an intuitive picture of the basic mechanisms behind dipole formation, allowing better design choices for future nanoscale systems such as single-atom catalysts.

The fabrication of surface structures at the atomic level is a technique with great potential. Scanning probe microscopy (SPM) is the method of choice in this context, since it permits the required atom-by-atom fabrication approach[1-7] as well as the structural and spectroscopic characterization of the fabricated structures. SPM therefore allows creating and studying artificial systems with properties that are exclusive to the nanoscale[8-12], like Majorana zero modes in low-dimensional systems[7,13-15].

An important aspect that is ubiquitous at the nanoscale is the electric potential field that exists around every atomic-scale object. These fields are caused by the interplay of the delocalized negative electric charge of electrons and the localized positive charges of atomic nuclei. While such potentials are difficult to measure with surface averaging techniques, the recently developed scanning quantum dot microscopy (SQDM) allows the imaging of the surface potentials $\Phi_s(\mathbf{r}_{||})$ associated with individual nanostructures as well as the quantification of the respective surface dipole moments $P_\perp$ perpendicular to the surface[16-19].

Understanding the electrostatic potentials around single adatoms and nanostructures is relevant for many applications. This is illustrated by the following examples, the list of which could be arbitrarily extended: In single-atom catalysts, which maximize catalytic efficiency while minimizing the amount of material needed[20-22], the charge distribution resulting from the interaction between the adatoms and the

[1]Peter Grünberg Institut (PGI-3), Forschungszentrum Jülich, 52425 Jülich, Germany. [2]Jülich Aachen Research Alliance (JARA), Fundamentals of Future Information Technology, 52425 Jülich, Germany. [3]Experimentalphysik IV A, RWTH Aachen University, Otto-Blumenthal-Straße, 52074 Aachen, Germany. [4]Control and Cyber-Physical Systems Laboratory, Technische Universität Darmstadt, 64277 Darmstadt, Germany. [5]II. Physikalisches Institut, Universität zu Köln, 50937 Köln, Germany. [6]Institute of Solid State Physics, NAWI Graz, Graz University of Technology, Petersgasse 16, 8010 Graz, Austria. [7]Department of Chemistry, University of Warwick, Gibbet Hill Road, Coventry, UK. [8]Department of Physics, University of Warwick, Gibbet Hill Road, Coventry, UK. ✉e-mail: c.wagner@fz-juelich.de

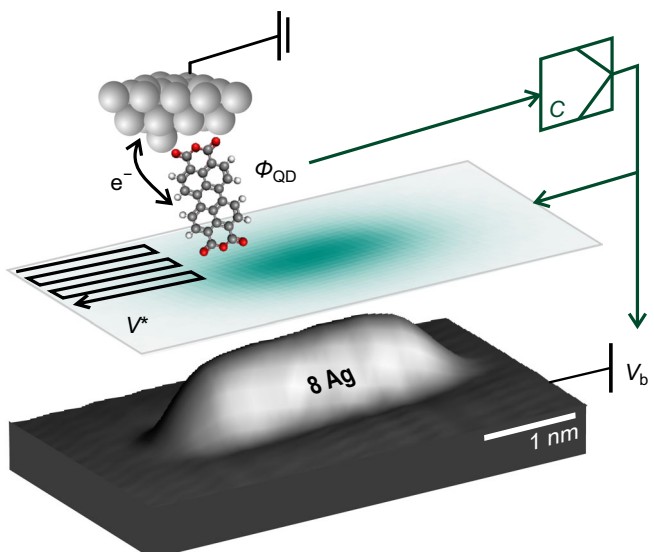

**Fig. 1 | Measurement principle of SQDM.** A single molecule, in this case PTCDA, is attached to the probe tip. It acts as a quantum dot (QD) that can be gated and charged with individual electrons by applying a bias voltage $V_b$ to the sample. To image the electrostatic potential $V^*$, the probe scans the sample (here, a chain of 8-Ag adatoms on Ag(111)) while changes in the surface potential beneath the tip are compensated by adjusting $V_b$ via a controller $C$, maintaining a constant QD potential $\Phi_{QD}$.

support is directly relevant to the catalytic activity[23]. Since homogeneous dispersion of single atoms is challenging, understanding how the charge distribution changes when multiple atoms are clustered is also relevant. Chains of adatoms, on the other hand, are very fundamental one-dimensional models to study collective effects and represent atomically well-defined 1D quantum wells that can be studied, for example, by tunnelling spectroscopy[4,24–30]. The charge distribution in such chains is relevant since it could modulate the potential landscape along the chain. Furthermore, the surface dipoles of isolated atoms or atomic chains are scattering centres for surface electrons, and the dipole strength could be a way to tune the intensity of the effects observed in quantum corrals and artificial lattices[12]. Finally, small atomic clusters form the apex of any scanning probe tip, and the Coulomb interaction between the tip apex and the sample is particularly important in applications where forces are measured with high sensitivity and spatial resolution[31,32]. Therefore, understanding the potential around such clusters is also relevant to interpret SPM measurements.

Here we use SQDM to study the surface potentials of single Ag and Au adatoms on the Ag(111) surface, as well as the collective effects that arise in chains and clusters of adatoms and that modify the respective surface dipoles. We observe opposite dipole polarities in the two systems and a qualitatively different behaviour of the per-atom dipole in atomic chains. We rationalize these results with elementary considerations pertaining to the electronic properties of the two atomic species. Our density functional theory (DFT) calculations allow a more detailed interpretation of the data and a quantitative comparison of the two model systems.

## Results

### Working principle of SQDM

Figure 1 illustrates the experimental setup which we use for the SQDM imaging of nanostructures. By means of molecular manipulation, a single PTCDA (3,4,9,10-perylene-tetracarboxylic dianhydride) molecule is attached to the metal tip of a combined non-contact atomic force / scanning tunnelling microscope (NC-AFM/STM)[33]. Since the molecular LUMO orbital hybridizes only weakly with tip states, PTCDA acts as a

quantum dot (QD) that can be gated by applying a bias voltage $V_b$ to the conductive sample[16,34]. The tip sample distance is typically on the order of 2–3 nm, such that there is no tunnelling current between the QD and the sample. At the critical gating voltages $V_b = V^+ > 0$ and $V^- < 0$ the charge state of the QD changes by one electron that either tunnels from the tip to the molecule at $V^+$ or vice versa at $V^-$. The change in the charge state is detected in the frequency shift ($\Delta f$) channel of the NC-AFM/STM, since the tip-sample force changes abruptly[19,34]. The electrostatic potential field of a nanostructure below the tip contributes to the gating, such that $V^+$ and $V^-$ change when the tip is moved laterally from an empty part of the surface to above a nanostructure. Tracking $V^+$ and $V^-$ in a two-pass approach as the SPM tip is scanned over the surface constitutes the basic imaging mechanism of SQDM[18].

From the primary measurands, $V^+$ and $V^-$, a representation $V^*(\mathbf{r}_{||})$ of the surface potential in the imaging plane, that is, at the height of the QD, can be calculated as

$$V^*(\mathbf{r}_{||}) = \frac{V^+(\mathbf{r}_{||}) - V^-(\mathbf{r}_{||})}{V_0^+ - V_0^-} V_0^- - V^-(\mathbf{r}_{||}), \quad (1)$$

where $V_0^+$ and $V_0^-$ are reference values acquired above a homogeneous, empty region of the surface[19]. The respective representation in the object plane, that is, the actual potential $\Phi_s$ on the surface (relative to the bare surface at which we define $\Phi_s = 0$) is related to $V^*$ in the form of a convolution with the point spread function (PSF) $\gamma^*$ as

$$V^*(\mathbf{r}_{||}) = \iint_{sample} \Phi_s(\mathbf{r}'_{||})\gamma^*(|\mathbf{r}_{||} - \mathbf{r}'_{||}|, z_t)\mathrm{d}^2\mathbf{r}'_{||}. \quad (2)$$

For the purpose of deconvolution, knowledge of $\gamma^*$ is required, which is mainly defined by the tip sample separation $z_t$ and, as a secondary effect, by the tip shape. In Refs. 18 and 19 we have shown that the assumption of a completely flat tip, which allows calculating the convolution kernel $\gamma^*$ from an infinite series of image charges, leads to satisfactory deconvolution results. Note that the interpretation of $V^*$ in SQDM and in high-resolution Kelvin Probe Force Microscopy (KPFM) images[35,36] is conceptually similar as both indicate changes in the surface potential.

Since the norm of $\gamma^*$ is 1, the 2D integrals of $\Phi_s$ and $V^*$ over a nanostructure have identical values as long as $\Phi_s$ and $V^*$ are zero on the boundaries of the respective integration areas $\mathcal{N}$ and $\mathcal{A}$, that is, for a nanostructure on an otherwise empty surface area. Since the surface potential can also be interpreted as a surface dipole density $\Pi_\perp = \epsilon_0\Phi_s$, the surface dipole $P_\perp$ of the nanostructure can be obtained in an integration as

$$P_\perp = \iint_{\mathcal{N}} \Pi_\perp(\mathbf{r}_{||})\mathrm{d}^2\mathbf{r}_{||} = \epsilon_0\iint_{\mathcal{N}} \Phi_s(\mathbf{r}_{||})\mathrm{d}^2\mathbf{r}_{||}$$
$$= \epsilon_0\iint_{\mathcal{A}} V^*(\mathbf{r}_{||})\mathrm{d}^2\mathbf{r}_{||}. \quad (3)$$

With these considerations, which are rigorously derived in Ref. 19, we determine the potential $\Phi_s$ on the surface by performing a measurement in the imaging plane above the surface (Eq. (1)), obtain the surface dipole from that image (Eq. (3)) and, if necessary, deconvolve the $V^*$ image to estimate the potential distribution on the surface (Eq. (2)).

In general, SQDM images show greatly enhanced resolution compared to KPFM images taken at similar tip-sample separations, because the QD is the only sensitive element and the tip and surface shield the electric fields[19]. At tip-sample separations of a few Å, KPFM and probe-particle measurements with passivated tips provide very high intra-molecular resolution[35–37], but the interpretation of these images is complicated as chemical interactions and structural relaxation set in

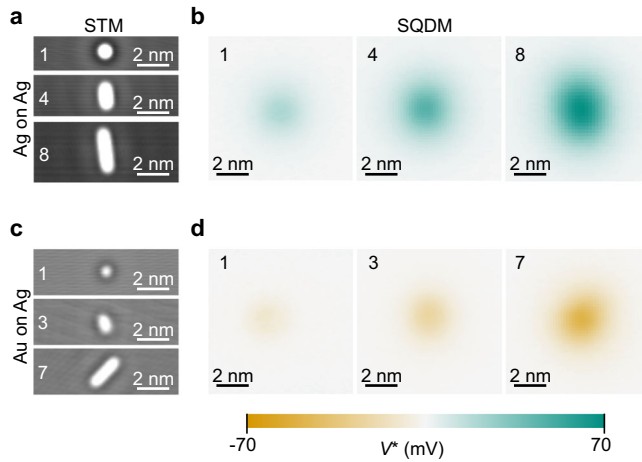

**Fig. 2 | STM topography and electrostatic potential images. a** Selected constant-current STM images of 1, 4, and 8 Ag adatom chains on Ag(111). **b** $V^*$ images of the chains in (**a**). **c** Constant-current STM images of 1, 3, 7 Au adatoms on Ag(111). **d** $V^*$ images of the chains in (**c**). The colour scale applies to all $V^*$ images shown.

and therefore the measurand does not provide a pure electrostatic potential signal anymore. While SQDM can also resolve intramolecular charge distributions[16], it does so in a non-invasive way at much larger tip-surface separations, precluding the modification of the system to be investigated by the probe tip and preserving quantitative interpretability as a convolution of the surface dipole density (Eq. (2)).

## Surface potentials and dipoles of adatom nanostructures

We performed SQDM measurements in a commercial qPlus-type NC-AFM/STM system operated at a base temperature of 5 K under ultra-high vacuum. To prepare our samples, we evaporated Ag or Au atoms, respectively, onto the Ag(111) sample placed inside the SPM. Subsequently, we fabricated chains and clusters by lateral manipulation of individual Ag or Au adatoms with the SPM tip, acquired $V^+$ and $V^-$ images, and computed the $V^*$ image via Eq. (1). The dipole $P_\perp$ is then obtained by integrating over the entire $V^*$ image (Eq. (3)). To avoid systematic errors, this image must therefore contain the entire contribution from either the chain or the cluster, but no contribution from any other nanostructure. To achieve this, we assembled the clusters or chains in areas far away from any defects, molecular islands, or step edges, and we recorded images of sufficient size so that the condition $V^* = 0$ is satisfied at the image boundary. The later condition requires image sizes between $15 \times 15$ nm$^2$ for adatoms and dimers, and $25 \times 25$ nm$^2$ for the 12-atom chain.

Cutouts of exemplary STM and $V^*$ images of individual Ag and Au adatoms and atomic chains are shown in Fig. 2. The $V^*$ images reveal that Ag adatoms and chains have positive electrostatic surface potential relative to the surrounding bare Ag(111) surface, while Au structures yield a negative surface potential. Note that positive dipoles point away from the surface, that is, their positive charge faces the vacuum. Interestingly, the Au atoms create (in absolute numbers) a weaker potential than Ag atoms which is, at first sight, surprising, since Ag on Ag is a chemically homogeneous situation, while Au on Ag is not. We will analyse and explain this aspect in detail below.

Even without further analysis it is clear from Fig. 2 that the magnitude of the electrostatic potentials increases with chain length. To quantify this effect, we have extracted the surface dipoles $P_\perp$ of isolated adatoms, adatom chains of various length and Ag clusters of 3 and 4 adatoms by integrating the respective $V^*$ images (Eq. (3)). The results are summarized in Fig. 3a where a linear increase of $P_\perp$ with chain length $N$ is revealed. The Ag dipole moments increase from $P_\perp(N = 1) = 0.66$ D for a single adatom, to

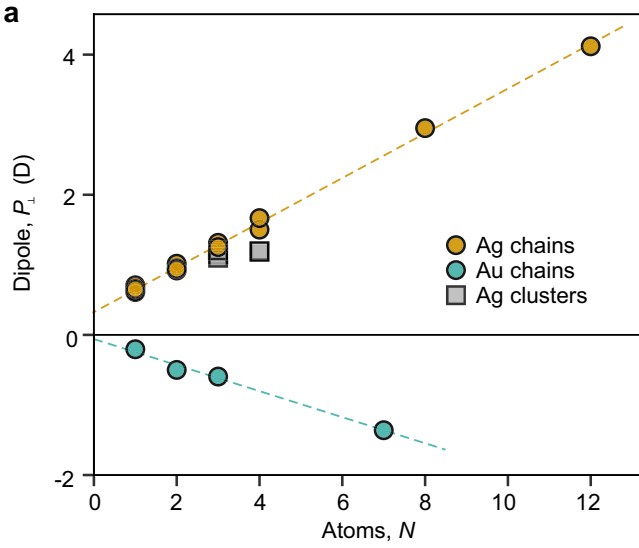

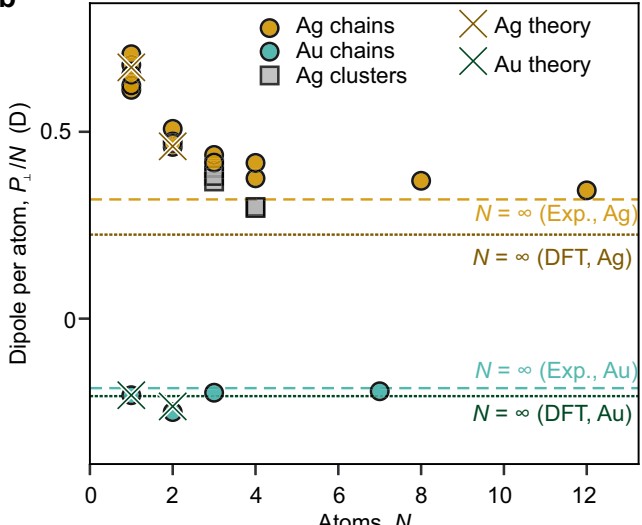

**Fig. 3 | Surface dipole moments. a** Measured total dipole moments $P_\perp$ of Ag adatom chains and clusters (dark yellow circles and grey squares), and Au adatom chains (teal circles). The uncertainty interval of our $P_\perp$ values can be assessed since we have fabricated and measured several Ag structures with $1 \leq N \leq 4$, the $P_\perp$ values of which largely overlap in the plot. **b** Dipole moment per atom $P_\perp/N$, for the data in (**a**). The crosses and dotted lines (brown and dark green) mark the values obtained from our DFT calculations for $N = 1, 2, \infty$, the dashed dark yellow and teal lines mark the respective asymptotic limits $N = \infty$ for the experimental data as obtained from the linear fits in (**a**).

$P_\perp(12) = 4.12$ D for the twelve-adatom chain. The dipole moments for the Au chains increase, correspondingly, from $P_\perp(1) = -0.20$ D to $P_\perp(7) = -1.36$ D. Since both trends are almost perfectly linear, we can determine the asymptotic value of the dipole per atom in an infinite chain by extrapolating the linear fit to $N \to \infty$ and dividing the resulting $P_\perp$ by $N$ (Fig. 3a). In this way, we find values of 0.32 D and $-0.19$ D for Ag and Au, respectively.

While the $P_\perp(N)$ relations of Ag and Au adatom chains have practically constant slopes, there is, in addition, a notable offset for the case of Ag: The increase in dipole obtained by adding the second, third, etc., adatom to the chain is considerably below the dipole of an isolated Ag adatom (Fig. 3b). We will discuss this effect later in the context of the DFT calculations.

Since the initial non-additivity of dipoles indicates that neighbouring atoms affect each other, it raises the question whether the per-

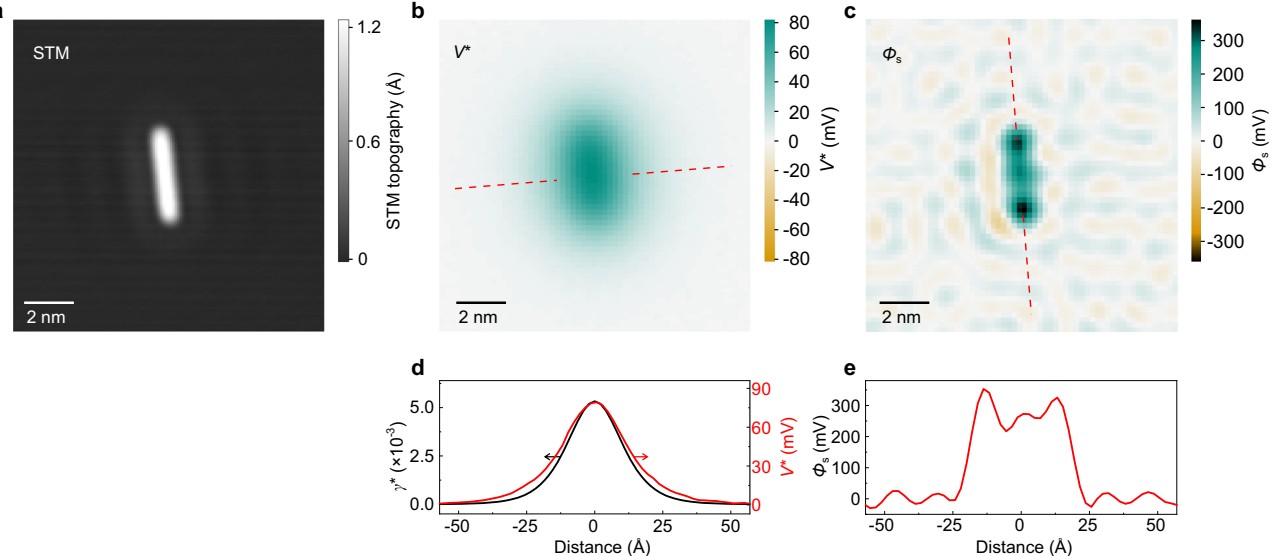

**Fig. 4 | Surface potential obtained by deconvolution. a** STM image of a 12-atom Ag chain. **b** SQDM $V^*$ image of the chain in (**a**). **c** Surface potential $\Phi_s$ obtained from deconvolving the $V^*$ image in (**b**) using $\gamma^*$ [18]. **d** Comparison of cross sections through the $V^*$ image (red) and through the used PSF $\gamma^*$ (black). **e** Cross section through $\Phi_s$ along the chain in (**c**) (red dashed). The patterns around the chain are deconvolution artefacts.

atom dipoles are homogeneously distributed along the entire adatom chain or whether some atoms contribute a larger while others contribute a smaller dipole to the total $P_\perp$ value. With SQDM, it is possible to address this question experimentally in surface potential images, as we now show for the 12-atom Ag chain.

The fact that $V^*$ images represent the electrostatic potential of the nanostructure above the surface at the height of the QD is reflected in the rather blurry appearance of the respective images in Fig. 2. Even for the 12-atom chain, the contrast is insufficient to discern any modulation along the chain (Fig. 4b). As explained above, it is, however, possible to retrieve the potential $\Phi_s$ close to the surface plane, which has a much higher lateral resolution than $V^*$, by image deconvolution with Eq. (2). For this purpose, we calculate the PSF $\gamma^*$ for the experimental tip-surface separation $z_t = 26$ Å in the approximation of a planar tip[19]. A comparison of cross sections through $\gamma^*$ and $V^*$ in Fig. 4d reveals that, perpendicular to the chain, the $V^*$ profile is approximately the same as the PSF, as it should be, apart from a small deviation that is expected because the PSF describes the $V^*$ contrast resulting from a single point in the object plane, while the surface potential of an atom is wider than that.

The result of the deconvolution procedure is shown in Fig. 4c. The $\Phi_s$ image clearly reveals the elongated structure of the chain. We note, however, that the dipole density is not distributed equally along the chain, but has maxima at both ends. This interpretation of the $\Phi_s$ contrast has to take the deconvolution artefacts into account, which are visible in the entire image. However, the peaks in $\Phi_s$ at both ends clearly exceed the amplitude of these artefacts, as can be seen in the line profile in Fig. 4e. The finding of higher dipole density at the chain ends indicates that the number of neighbours in the immediate vicinity of an Ag adatom influences its surface dipole. This effect is further confirmed by comparing the total dipole moments of clusters and chains of equal adatom count (Fig. 3a). In the three- and four-adatom clusters, all atoms have at least two neighbours, whereas in the corresponding chains two atoms have only a single neighbour. Consequently, the dipole of the clusters is expected to be lower than that of the chains, which is indeed observed experimentally. In the following section, we will study this aspect of neighbourship relations in more detail with the help of DFT calculations and subsequently sketch out an intuitive explanation for the observed phenomenology.

## DFT calculations

The accuracy of our measured $P_\perp$ values establishes them as a benchmark for ab initio calculations. Unlike most benchmarks that test either formation, adsorption or state energies, here the focus is on the total charge density distribution, as this is what determines the surface dipoles. In previous work, we showed for PTCDA molecules on Ag(111) that an accurate DFT-based prediction of $P_\perp$ at a molecule-metal interface is non-trivial and can be subject to substantial deviations from the experimental value[18]. Here, we apply DFT using the Perdew-Burke-Ernzerhof (PBE) functional[38] to single Au and Ag adatoms, dimers and infinite chains on Ag(111). The use of periodic boundary conditions allows studying the asymptotic limit $N \to \infty$ which is not accessible in experiment.

The dipole is obtained from the charge density $\rho$ as

$$P_\perp = \int z\rho_{xy}(z)\mathrm{d}z, \tag{4}$$

with

$$\rho_{xy}(z) = \int\int \rho(x,y,z)\mathrm{d}x\mathrm{d}y. \tag{5}$$

The calculated surface dipoles for one and two adatoms and the infinite chain in Fig. 3b agree very well with the measured $P_\perp$ values for both species of adatoms, suggesting that the underlying physics of the metal-metal interaction is well captured in the calculations (for data on convergence see Supplementary Fig. 1). The only significant discrepancy is found for the infinite Ag chain, where the theoretically predicted dipole per atom is 0.09 D too small, highlighting the utility of our data for benchmarking of ab-initio methods.

To reach a deeper understanding of the observed surface dipoles, we calculated the charge density difference

$$\Delta\rho = \rho_{sys} - (\rho_{ads} + \rho_{subst}) \tag{6}$$

which results from the adsorption of single Ag and Au adatoms (Fig. 5), where $\rho_{sys}$ is the calculated charge density of the combined system, while $\rho_{ads}$ and $\rho_{subst}$, respectively, refer to the densities of adsorbate and substrate separately. The isosurface plots of the charge density

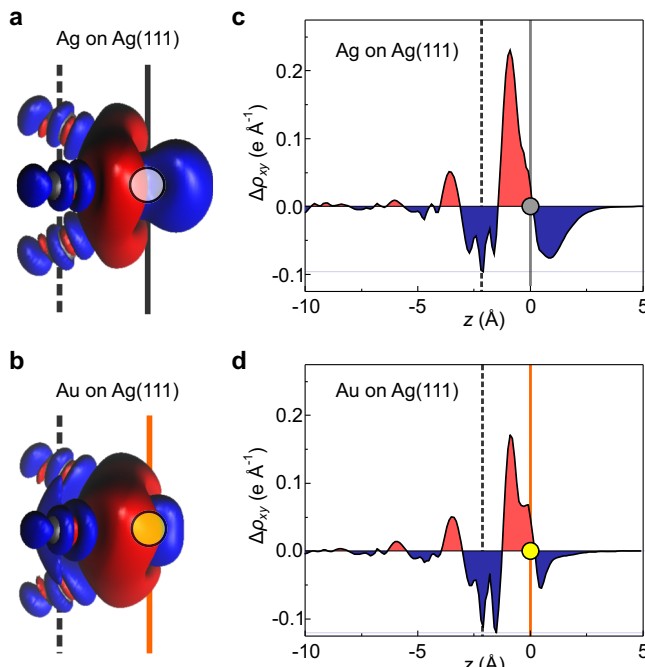

**Fig. 5 | Charge density differences of Ag and Au adatoms. a, b** Isosurface plots of $\Delta\rho(\mathbf{r}) = \pm 0.007$ e Å$^{-3}$ for a single Ag or Au adatom, respectively. Charge accumulation is shown in red, depletion in blue. Indicated are the $z$ height of the Ag surface atoms (grey, dashed) and of the adatom at $z = 0$ (solid). **c, d** Plots of the $xy$-integrated charge density difference $\Delta\rho_{xy}(z)$ for a single Ag or Au adatom (indicated as grey or yellow circle respectively).

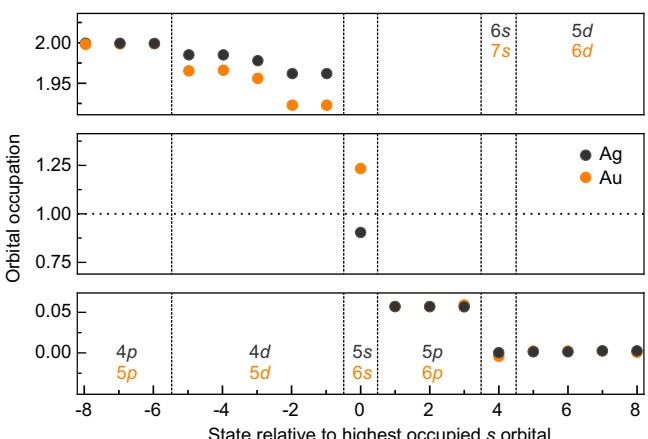

**Fig. 6 | Orbital occupation.** Occupation of several orbitals of single Ag and Au adatoms on Ag(111). Vertical dashed lines separate subshells. The horizontal dotted line indicates half occupation. Note that $y$-axis scaling varies between panels.

difference $\Delta\rho = 0.007$ e Å$^{-3}$ in panels (a) and (b) illustrate how the charge densities of the adatoms and the surface are modified as the two are brought into contact. It turns out that there is a pronounced region of charge depletion above the Ag adatom, while this depletion region is more confined and largely embedded into the accumulation region in case of Au. The $xy$-integrated densities $\Delta\rho_{xy}(z)$ in Fig. 5c, d allow a more quantitative assessment of the respective charge rearrangements. As expected, they show a clear separation of accumulation and depletion along $z$ for the Ag adatom (Fig. 5c), while the two contributions cancel each other to some degree in the integrated $\Delta\rho_{xy}(z)$, $z \geq 0$ for Au (Fig. 5d). Since the adsorption heights of Ag and Au adatoms are virtually identical in the DFT calculations, height variations can be ruled out as a contributing factor.

To check whether the observed differences in the charge rearrangement originate from the involvement of different atomic orbitals in the adsorption process, we calculated the occupation of the respective (projected) $s$ and $d$ orbitals (Fig. 6). In vacuum, Ag and Au possess similar electronic configurations, namely a filled $d$ shell and a single $s$ electron. The calculations reveal a qualitative difference in the occupation of the $s$ orbitals, where the $5s$ orbital occupation of Ag drops below 1 upon adsorption, while the corresponding $6s$ occupation of Au rises above 1. In contrast, a slight depletion of the $d$ orbitals and a slight charge accumulation in the $p$ orbitals is observed for both atomic species. Summarizing, the Ag adatom loses more charge from its $s$ and $d$ orbitals than it receives into $p$, while Au receives more charge into its $s$ and $p$ orbitals than it loses from its $d$ orbitals. Thus, we can tentatively associate the extended depletion region above the Ag adatom with the reduced occupation of the $s$ and $d$ orbitals, while the more confined depletion above the Au adatom arises from the counteracting increase and decrease, respectively, of the $s$ and $p$ versus the $d$ orbital occupations.

Finally, we investigated how the formation of a dimer, in other words the binding between two adatoms, affects the charge density rearrangement. To this end, we calculated

$$\Delta\Delta\rho(x,y,z) = \Delta\rho_{\text{dimer}} - (\Delta\rho_{\text{left}} + \Delta\rho_{\text{right}}), \qquad (7)$$

that is, the change in the charge density that occurs when two already adsorbed adatoms ("left" and "right") are brought into contact on the surface. In Fig. 7 we compare this change $\Delta\Delta\rho$ with the change $\Delta\rho$ that occurs for two isolated (i.e., gas-phase) Ag or Au atoms at an identical separation. It shows that the overall pattern of the charge redistribution is the same for isolated atoms and adatoms and leads to a charge accumulation along the dimer axis (the blue depletion regions close to the two atoms are ring-shaped). However, the adatom case differs in two distinct ways. First, the depletion region in the centre of the dimer is strongly modified, forming a teardrop shape slightly below the adsorption plane of the Ag and Au adatoms. To highlight the second aspect, we compare the adsorbed dimers with the single adatoms discussed above. Comparing the $xy$-integrated densities $\Delta\Delta\rho_{xy}(z)$ in Fig. 7c, d with the corresponding $\Delta\rho$ curves for the single adatom (grey), it becomes clear that the formation of the Ag dimer partially reverses the charge rearrangement effects that occur during the adsorption of the isolated atoms: where the grey $\Delta\rho_{xy}(z)$ curve shows charge accumulation, the $\Delta\Delta\rho_{xy}(z)$ curve shows (weaker) depletion and vice versa. For the Au dimer, the same effect can be observed, but a second trend is superimposed, which can be roughly described as a relocation of charge from above the adatoms to the plane of the adatoms (Fig. 7d), creating a positive dipole contribution.

In summary, both dimers show similar charge rearrangement patterns (teardrop) that cause a negative differential dipole contribution; however, in the case of Au, this is counteracted by a second rearrangement mechanism that is responsible for a positive contribution. This explains the measured reduction of the per-atom dipole for Ag dimers and the constant per-atom dipole for Au dimers as shown in Fig. 3.

## Discussion

The opposite signs of the Ag and Au adatom dipoles and the differences in the calculated charge distribution and orbital occupation ask for an intuitive explanation. While the calculated dipoles accurately reproduce the measured values, an intuitive understanding could, beyond that, allow rational design choices for other surface-adatom systems. In our discussion we first address and compare the properties of single adatoms and subsequently turn to the dimers.

The differences in orbital occupation and $\Delta\rho$ distribution in Figs. 5, 6 can be explained by a combination of three different effects of very general nature.

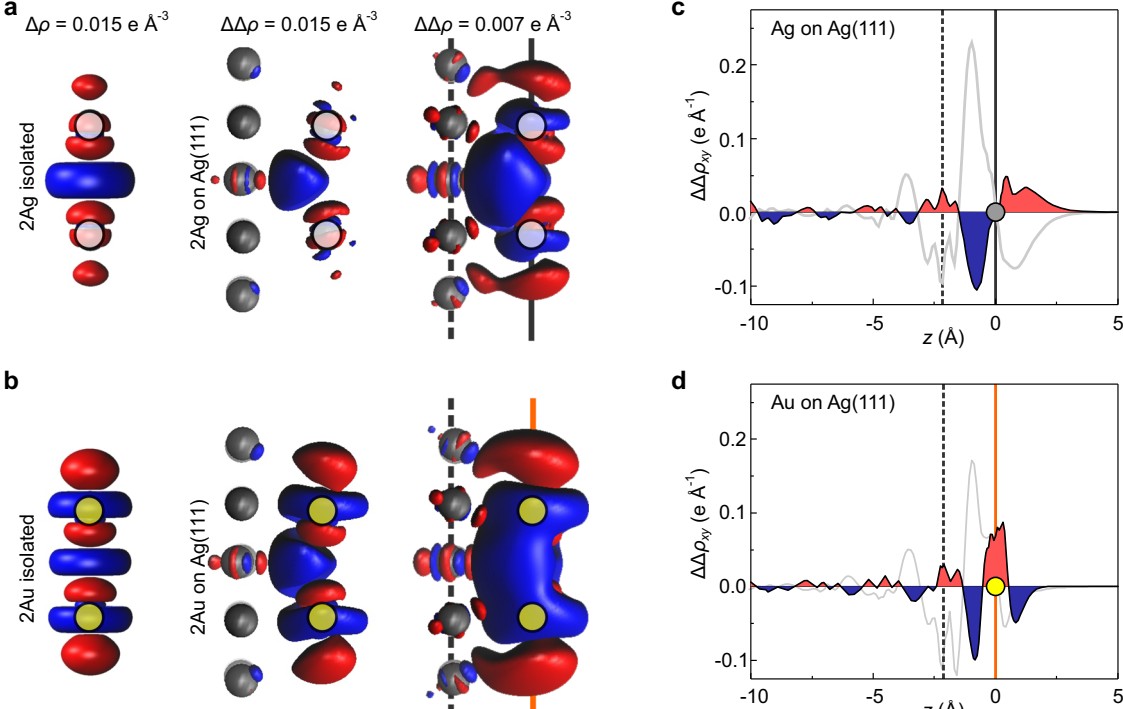

**Fig. 7 | Charge density difference for dimer formation. a, b** Isosurface plots for isolated Ag and Au dimers ($\Delta\rho(\mathbf{r}) = \pm 0.015$ e Å$^{-3}$, left) and for Ag and Au adatom dimer on the Ag(111) surface ($\Delta\Delta\rho(\mathbf{r}) = \pm 0.015$ e Å$^{-3}$ and $\pm 0.007$ e Å$^{-3}$, centre, right). Charge accumulation is shown in red, depletion in blue. Vertical lines indicate the $z$ height of the Ag surface atoms (grey, dashed) and of the adatom (solid grey (Ag) or orange (Au)). Adatom positions are indicated by filled grey and yellow circles. **c, d** Plots of the $xy$-integrated charge density difference $\Delta\Delta\rho_{xy}(z)$ for the formation of Ag or Au dimers on the surface. The $\Delta\rho_{xy}(z)$ curves for single adatoms from Fig. 5 are shown in light grey. Surface and adatom positions are indicated by vertical lines.

(1) The free electrons of the conductive substrate screen nearby charges (image charge effect), thereby lowering their potential energy and attracting them towards the surface. This effect is strong enough to deform the charge density of the loosely bound outer electron(s), while the ion core largely retains its spherical symmetry such that the adatom becomes polarized. This deformation, which is present even when there is no overlap between substrate and adsorbate charge densities, causes a charge depletion above the adatom and an accumulation below, leading to the polarization of the adatom electron density in the presence of the surface.

(2) The second aspect to be considered here is the charge rearrangement associated with atomic-scale surface roughness. This effect, described by Smoluchowski[39], can be explained by the tendency of the charge density to avoid the sharp contours of an atomic-scale surface roughness and instead smooth out, thereby lowering its kinetic energy. This process levels the slopes around a protrusion with excess charge while creating charge depletion above its centre, similar to the polarization in (1). The combination of both effects can thus explain the depletion of the Ag 5s orbital, the positive dipole and the charge density difference $\Delta\rho$ calculated for Ag (Fig. 5a). However, for the Au adatom on Ag(111), which is not a homogeneous case as far as the atomic species are concerned, a third aspect has to be considered.

(3) The third aspect is a substrate-adsorbate charge transfer. Our DFT calculations reveal a net charge transfer into the 6s orbital of the Au adatom (Fig. 6). This can be understood by taking into account the higher ionization energy and higher electron affinity of Au atoms compared to Ag. Removing an electron from the Au 6s orbital requires 1.6 eV more energy than removing the 5s electron of Ag. In contrast, removing a 5d electron of Au requires 1.4 eV less energy than removing a 4d electron of Ag[40]. Finally, the electron affinity of Au is 1.0 eV higher than that of Ag[41]. Thus, it is not surprising that the 5d states of Au adatoms loose more charge to the surface than the 4d states of Ag

adatoms and that, conversely, charge is transferred from the surface into the Au adatom's 6s orbital while the 5s orbital of Ag loses charge to the surface (Fig. 6). Importantly, the substrate-adsorbate charge transfer can, in principle, take both directions, such that the resulting surface dipole contribution can be either positive or negative. Since Au receives charge, the respective dipole is negative and over-compensates the contributions from (1) and (2), which are present for both Ag and Au adatoms, thus explaining the measured small negative dipole value (Fig. 3).

The step from a single adatom to a dimer has markedly different consequences for Ag and Au adatoms. While the (absolute) surface dipole of the dimer is larger than that of a single adatom in both cases, the dipole per atom decreases for Ag, but stays constant for Au adatoms. Our experimental data show that this trend is continued also for longer chains (Fig. 3b). To understand both effects, we need to consider the consequences of joining two adatoms on the contributions (1)-(3) that were identified as responsible for the charge density rearrangement in the first place.

The strength of both the polarization effect (1) and the Smoluchowski smoothing (2) will be attenuated by the formation of a dimer. The deformation of the charge density upon screening is reduced, because the presence of two parallel dipoles naturally causes depolarization. The dimer formation likewise reduces the charge accumulation between the adatoms resulting from Smoluchowski smoothing, since this region loses its properties as a kink in the surface topography and the flattened out charge distributions right between the two adatoms moreover experience some Pauli repulsion. In combination, both effects can explain the teardrop-shaped region of (differential) charge depletion between the adatoms in $\Delta\Delta\rho$ (Fig. 7) which is, in fact, primarily a region of reduced charge accumulation. Since (1) and (2) are present in Ag and Au adatoms it is not surprising that the respective teardrop feature in $\Delta\Delta\rho$ is present for both species as well

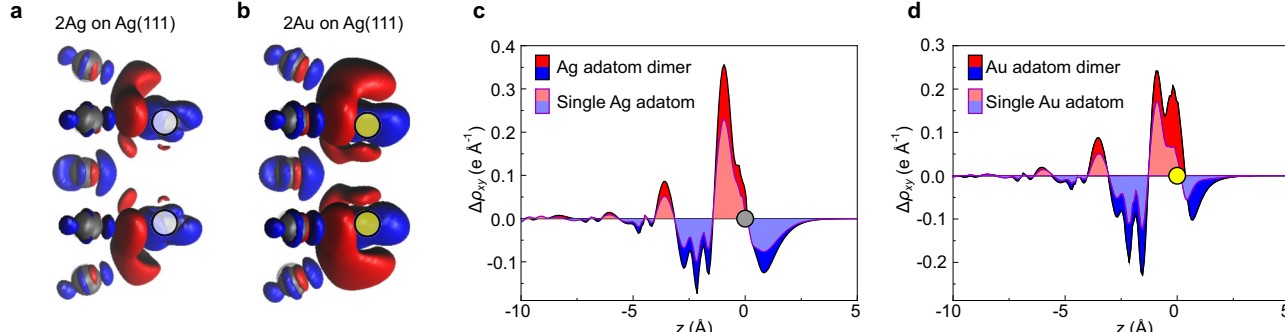

**Fig. 8 | Charge density difference for dimer adsorption. a, b** Isosurface plots of $\Delta\rho(\mathbf{r}) = \pm 0.02$ e Å$^{-3}$ for the adsorption of a Ag or Au adatom dimer (as a whole) onto Ag(111). Adatom positions are indicated by filled grey and yellow circles. Charge accumulation is shown in red, depletion in blue. **c, d** Comparison of the *xy*-integrated charge density difference $\Delta\rho_{xy}(z)$ for single adatoms (light colours) and adatom dimers (dark colours) of Ag and Au.

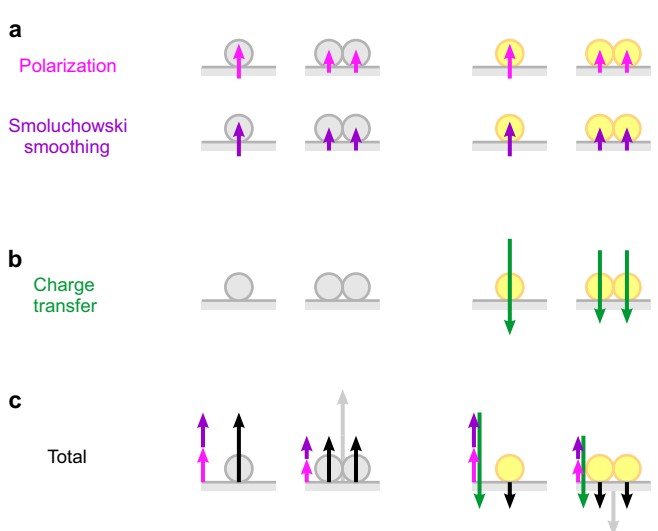

**Fig. 9 | Schematic summary of dipole contributions. a** The per-atom dipole contributions from polarization (pink) and Smoluchowski smoothing (purple) are positive for Ag (grey) and Au (yellow) and decrease upon dimer formation. **b** The charge transfer (green) which is only present for Au adatoms is negative and likewise decreases when a dimer forms. **c** As a result, the total (absolute) per-atom dipole (black) of Ag adatoms decreases upon dimerization, while it remains constant for Au adatoms. Grey arrows represent the dipole moments of the dimers. Note that all arrow lengths only visualize trends without being quantitative.

(Fig. 7a and b). This interpretation is consistent with the observation of stronger surface potentials at both ends of an adatom chain (Fig. 4c), where the adatoms have only one instead of two neighbours.

The effect of dimer formation on the substrate-adsorbate charge transfer (3) can be rationalized by considering the electronic configuration of the Au dimer. In the case of the single adatom, we found a charge transfer into the 6*s* orbital (Fig. 6), which is favourable since the respective orbital is singly occupied. In the dimer, however, the two 6*s* electrons form a fully occupied bonding orbital and an empty antibonding orbital. Regardless of the effect of hybridization with Ag(111) states, the charge transfer into the respective orbitals becomes less favourable, because the lower-energy bonding orbital is occupied and the empty antibonding orbital is higher in energy. In the isolated (gasphase) Au dimer (Fig. 7b, left) this leads to a reduction of the electron affinity by 0.37 eV compared to a single Au atom[41,42]. Thus, we would expect a reduction of the charge transfer (per atom) upon the formation of an adatom dimer. This compensates the reduction in the contributions of (1) and (2) discussed above and leads to the observed constant per-atom dipole of Au adatoms in chains of various lengths. The assumption that charge is, in fact, transferred into the

antibonding orbital of the adatom dimer is corroborated by the observation of regions of strong charge accumulation at the far ends of the dimer. This is shown in Fig. 8 where we compare $\Delta\rho$ (Eq. (6)) for Ag and Au dimers upon adsorption (as a whole). Finally, Fig. 9 illustrates the three contributions to the total dipole discussed here and summarizes how they vary between adatoms and dimers.

If there is substantial charge transfer, the general tendency for effects (1), (2), and (3) to weaken upon chain formation can lead to peculiarities such as a reversal of dipole polarity upon dimer formation. In exploratory DFT calculations we indeed find this effect for Pd adatoms on Ag(111) with per-atom dipoles of -0.06 D, 0.07 D, and 0.19 D for adatom, dimer, and chain, respectively.

It should be noted that, in the limit of a full layer of Ag or Au adatoms, the per-atom dipole of the Ag adlayer will naturally become zero as there is no difference between the adlayer and the bare surface, while the Au adatoms will keep a non-zero dipole per atom which will eventually lead to the difference in the work functions of Au and Ag as more Au layers are added. Hence, even beyond the microscopic arguments given here, it is clear that the adatom dipoles have to reach a zero or non-zero value for Ag and Au, respectively, in the asymptotic case of a closed layer.

In summary, SQDM allowed us to follow the evolution of the electrostatic properties of metallic nanostructures with atomic size and shape. We performed quantitative imaging of the electrostatic potential of chains and clusters of Au and Ag adatoms on the Ag(111) surface, extracted the surface dipoles $P_\perp$ of these nanostructures and compared them with DFT calculations. A deeper analysis of the calculated charge density differences allowed to determine which effects in combination provide an intuitive explanation for the observed variations of $P_\perp$ with adatom species and chain length. These explanations, which link dipoles to elementary effects and atomic properties, can serve as guidelines for the future design of functional nanoscale surface modifications or devices.

## Methods
### Experiment
The experiments were performed in a low-temperature NC-AFM/STM (Createc GmbH) in ultra-high vacuum at 6 K. We prepared the Ag(111) surface by repeated cycles of Ar$^+$ sputtering with an ion beam energy of 1 keV and annealing at 850 K. The substrate was held at room temperature during the molecule deposition. We deposited PTCDA (3,4,9,10-perylene-tetracarboxylic dianhydride) by sublimation from a UHV evaporator, heated to 580 K.

The apex of the tip was prepared and reshaped in the STM at low temperature by applying voltage pulses and indenting the tip into the clean Ag surface. We checked the quality and sharpness of the tip by imaging a single adatom until we observed a circular shape and detected a small $\Delta f(z)$ shift ($\approx$ 2-4 Hz) in the AFM signal.

We deposited individual silver and gold atoms onto the PTCDA/Ag(111) surface by controlled thermal sublimation from custom-made evaporators and fabricated chains and clusters of different sizes by lateral single-atom manipulation[1,43]. Single Ag adatom relocation was achievable at tunnelling resistances of $10^5$ Ω, while the corresponding value was $6.7 \times 10^4$ Ω for Au.

The final SQDM images of the nanostructures require recording two intermediate maps, measured at $V^+$ and $V^-$, of the same sample area to record charging and discharging events on the QD. A controller tracks a specific $\Delta f$ value at the slope of each peak[44,45], which provides a voltage $\Delta V$ that is added to the applied bias $V_b$ to track and compensate the changes in $V^+$ or $V^-$ as the tip scans over the nanostructures.

We have fabricated and measured a total of 23 chains or clusters (for Ag: 5 × adatom, 3 × dimer, 2 × 3-atom chain, 3 × 3-atom cluster, 2 × 4-atom chain, 2 × 4-atom cluster, 1 × 8-atom chain, 1 × 12-atom chain; for Au: 1 × adatom, 1 × dimer, 1 × 3-atom chain, 1 × 7-atom chain).

## Data processing
Each surface potential image consists of two maps of the quantum dot charging event denoted $\Delta V^+(x, y)$ and $\Delta V^-(x, y)$, which are related to the bias voltage applied to the junction, with positive and negative values, respectively.

In the analysis of each $\Delta V^{+,-}(x, y)$ map, we averaged the data recorded in the forward and backward movement of the tip. We applied a tilt correction to correct drift effects. The tilt corrections were based on individual linear fits for both scan directions. To avoid the influence of the nanostructure at the centre of the image, we used only the left, right, top and bottom edges of the image for the fit. To calculate the $V^*$ image, we measured the reference values $\Delta V_0^{+,-}$ in the four corners of the image and averaged the value. Since a potential misalignment due to lateral drift can occur in two-pass imaging, we precisely aligned the centres of the nanostructure in the $V^+$ and $V^-$ images, if required. Finally, we calculated $V^*$ using Eq. (1).

## Density functional theory calculations
The DFT calculations were performed using a 5 × 5 Ag(111) surface slab with single adatoms, 2 adatoms (dimers), and complete rows (5 atoms with periodic boundary conditions). The substrate is represented with 6 layers of which the bottom two are frozen in their bulk configurations. All calculations were performed with the numeric atomic orbital code FHI-aims[46] using the PBE exchange-correlation functional[38] including a long-range dispersion correction based on the vdW$^{surf}$ method[47]. We used a 6 × 6 × 1 k-point grid, scalar relativistic corrections and a default tight basis set for all calculations. Geometries of all reported structures were optimized until a maximum force threshold of 0.01 eVÅ$^{-1}$ was achieved. All calculations were performed with a dipole correction to ensure an accurate representation of the electrostatic potential. The reported dipole moments were calculated based on the change in work function associated with adsorption of adatoms, dimers, or one-dimensional metallic chains.

## Reporting summary
Further information on research design is available in the Nature Portfolio Reporting Summary linked to this article.

# Data availability
The data that support the findings of this study are available from the corresponding author upon request. All reported structures including input and output files have been deposited on the NOMAD electronic structure repository[48]. The experimental images and data are available in the central institutional repository for research data of Forschungszentrum Jülich, Jülich DATA[49].

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

## Acknowledgements

RJM acknowledges funding through a UKRI Future Leaders Fellowship (MR/S016023/1). High-performance computing resources were provided via the Scientific Computing Research Technology Platform of the University of Warwick. RJM and OTH collaborated during a sabbatical visit supported by the Warwick Institute of Advanced Study Fernandes Fellowship programme. OTH and RJM thank the sponsor, Ruis Fernandes, for his generosity. FST acknowledges financial support by the Bavarian Ministry of Economic Affairs, Regional Development, and Energy (grant allocation no. 07 02/686 58/1/21 1/22 2/23). RB, JMG, PL, MK, MM, and CW acknowledge funding through the European Research Council (ERC-StG 757634 "CM3").

## Author contributions

R.T., F.S.T. and C.W. conceived and designed this research. R.B., P.L. and M.K. performed the experiments. R.B., J.M.G. and C.W. analysed the data. M.M. and R.F. designed and provided the SQDM feedback controller. O.T.H. and R.J.M. conducted the DFT simulations with contributions from H.H.A. O.T.H., R.J.M., F.S.T. and C.W. interpreted the data and wrote the paper with contributions from R.B. and J.M.G.

## Funding

## Competing interests

The authors declare no competing interests.
