## [Peer Review File · Nature Communications]

Electrostatic potentials of atomic nanostructures at metal surfaces quantified by scanning quantum dot microscopyREVIEWER COMMENTS

Reviewer #1 (Remarks to the Author):

The manuscript entitled "The electrostatic potential of atomic nanostructures on a metal surface" by R. Bolat et al. investigates the evolution of the electric potential in short gold and silver atomic chains built up atom by atom on Ag(111) using a scanning tunneling microscope. The authors also employed a technique that they have developed, the scanning quantum dot microscopy (SQDM), to quantify the surface potentials and extract vertical dipole moments of each structure. These experimental dipole moments are then compared with the charge density and orbital occupations obtained by density functional theory calculations (DFT). The authors attempt to rationalize the evolution of dipole moments by considering three effects being (1) the screening of the metal free electrons leading to the polarization of the surface/adatom electron density, (2) the Smoluchowski effect and (3) substrate-adsorbate charge transfer. When dimers are formed, the charge transfer is impacted by the formation of bonding/antibonding orbitals, that reduces the contributions of (1) and (2) leading to a constant per-atom dipole of Au adatoms in chains of various lengths. Although this mechanism described by the authors is not quantitative, the explanation is plausible to explain the difference of dipole moments between Au and Ag adatoms and their evolution with chain lengths.

To me, the main finding is the observation of opposite dipole polarities for Au and Ag nanostructures on Ag(111), which can only be obtained using the SQDM technique. The DFT calculations nicely supports these experimental observations by explaining how charge transfers modifies the dipole magnitude. The manuscript is well-written and the experimental/theoretical data are accurately described. From a general point of view, this is an interesting application of the SQDM technique to nanoscale science which, to my opinion, shows the potential of the technique and deserves publication. Therefore, I recommend the publication of the present manuscript in Nature Communication after minor revisions.

Here are few comments :

1- In the introduction, the authors state l 54: "In addition, chains of magnetic adatoms created on the surface of a superconductor are candidates for observing Majorana zero modes [7, 13, 14, 31, 32] the quantum properties of which could be exploited in future devices."

I am not sure this sentence of the introduction is really relevant for the paper. The authors have neither characterized magnetic atoms nor on a superconducting surface. Besides, the formation of Majorana zero modes in atomic chains is bound to the emergence of Yu-Shiba-Rusinov states, that primarily depends on the magnetic exchange interaction of the atom with the superconductor. For sure, dipole moments and charge transfers of adatoms upon adsorption on the superconductor will play a role to the YSR band diagram . If the authors wish to keep this statement, they should develop their thoughts and be more specific.

Following this comment, I would suggest the authors to discuss the important of dipole moments in atomic structures made atom-by-atom with the aim of confining surface electrons of metals. Quantifying dipole moment per atoms and the underlying mechanism of their formation seems to me more interesting in this context (examples can be found here: <https://www.nature.com/articles/s42254-019-0108-5>).

2- l149: "Each nanostructure was created in a sufficiently large empty surface region, devoid of any surface or sub-surface defects, such that $V =! 0$ on the image boundary A (compare Fig. 4), as required for the application of Eq. 3."

There is a strange typo and the sentence is awkward. The authors should correct the sentence.

Reviewer #2 (Remarks to the Author):

The authors of this manuscript report the results of a combined experimental and theoretical study into the electrostatic potential of atomically well-defined nanostructures on surfaces. In particular, it is shown that scanning quantum dot microscopy (SQDM) can be used to quantitatively measure the dipole moment of atoms, and chains thereof. The experimental findings are reproduced by DFT calculations, and it is claimed that the experimental data can be used as a benchmark for theoretical calculations.

Spatial variations in the charge distributions play a crucial role in various fields, ranging from chemical reactivity/catalysis, to simulating matter with artificial lattices. Consequently, I believe the results presented here are of interest to a broad audience.

The manuscript is clearly written (a general introduction into SQDM is much appreciated, and I believe useful to the non-specialist reader) and the arguments are easy to follow. The claims in the paper are supported by the data presented and the methodology is sound (I have some comments about the theoretical aspects, see below). The methods section, together with the indicated references, provide sufficient detail for others to reproduce the results.

I have a few points which might help improve the manuscript.

- The agreement between experiment and theory is impressive (c.f. Fig.3b). However, to better substantiate the claim that the experimental results enable them to be used as a benchmark for theoretical calculations, I believe it is crucial to explicitly show that the theoretical results are converged with respect to e.g. k-point mesh and basis set for the quantity of interest (dipole moment). Probably, the authors have already performed these checks. It would be good to add those checks to the supplementary information. Furthermore, DFT results are typically affected by the (type of) exchange-correction functional. To further substantiate the claim of acting as a benchmark, it would be good to show a case where the converged DFT results do not match the experimental data.

- Inhomogeneous charge distributions, as well as the electrostatic force field of individual molecules have been visualized with Kelvin Probe Force Microscopy and Atomic Force Microscopy with charged tips before. It would be beneficial if the authors could include a paragraph describing the merits/limitations of each approach.

Minor points:

- On page 3, the authors write: 'Each nanostructure was created in a sufficiently large empty surface region, devoid of any surface or sub-surface defects.' What does 'sufficiently large empty surface region' mean concretely?

- Some of the affiliations appear identical, e.g. 2 and 9, 3 and 10.

Once these points are addressed, I recommend to publish this manuscript.

Reviewer #3 (Remarks to the Author):

The authors present a combined experimental and theoretical study of the electrostatic potential emanating from single atoms, bottom up built atomic chains and clusters using a scanning probe based method. As correctly argued by the authors, the presence of electrostatic potentials at surfaces plays a crucial role, e.g., in catalysis, but are difficult to determine quantitatively on a nanometer scale. The paper is very well written and intelligible organized. Moreover, it was interesting and enjoyable to read.

In the past, the authors developed and established an ingenious approach to measure the electrostatic potential by attaching a molecule to the tip apex of an STM/AFM tip. Its charge state is then deliberately manipulated to map out the local electrostatic potential. These results can be readily compared to DFT calculations, which allow to obtain the electrostatic surface potential from the total charge density. These results can be used to get an intuitive understanding in terms of electrostatic surface dipoles stemming from polarization effects, Smoluchowski smoothing and charge transfer.

In conclusion, this is an excellent piece of work and thus should be published as it is with only minor changes (see further below). The content of the paper is very useful for experimentalists as well as theoreticians. The latter can actually use such data for benchmarking the DFT calculations to predict or reproduce surface dipole moments, while the former might employ the method to get a better understanding of local variations of the electrostatic potential due to the presence of surface features like atoms, clusters, chains or step edges.

Having said this, I would like to add some questions and comments. They all revolve around the issue of lateral resolution, distance dependence and sensitivity and the comparison between SQDM and KPFM-based LCPD measurements.

1. According to Fig. 9, the dipole moments in clusters or chains should be localized at the individual atoms. However, atomic resolution is not obtained, only a slight modulation of the surface potential map on the 12-atom chain can be seen in Fig. 4c, i.e., maxima at the chain ends and the central part. What limits the lateral resolution of SQDM? Note that I am aware of the data in Ref. [18], where submolecular features are visible on larger molecules, but not with atomic resolution and not with the intramolecular resolution of KPFM-based LCPD measurement shown, e.g., in *Nature Nanotechnology* 7, 227 (2012).

2. Only laterally resolved images are shown. What about the height dependence of the signal? Are height-dependent data difficult to obtain or hard to evaluate?

3. V^* has been also used to represent the LCPD in high resolution KPFM images on single molecules, as e.g., in the before-mentioned publication *Nature Nanotechnology* 7, 227 (2012). The difference should be mentioned in the text to avoid confusion, because this quantity has been related to the electrostatic surface potential as well, albeit only qualitatively.

Optional issues that could be addressed by the authors and might be insightful for the reader:

4. The authors might want to discuss the differences between their method and LCPD data on molecules with intramolecular resolution (no atomic resolution, but quantitative vs. atomic resolution, but only a qualitative comparison with DFT possible) and why the lateral resolution of the latter method seems to be better (cf. point 1).

5. On single molecules subatomic atomic charge redistributions were detected using KPFM-based LCPD measurements, e.g., sigma-hole imaging and detection of the quadrupole moment of CO [both discussed in *Science* 374, 863 (2021)] and pi-hole imaging [*Nature Communications* 14, 4954 (2023)]. Is it possible to obtain similar sensitivity while keeping the quantitative accuracy with the method presented by the authors?

Response to reviewers

Reviewer #1

The manuscript entitled “The electrostatic potential of atomic nanostructures on a metal surface” by R. Bolat et al. investigates the evolution of the electric potential in short gold and silver atomic chains built up atom by atom on Ag(111) using a scanning tunneling microscope. The authors also employed a technique that they have developed, the scanning quantum dot microscopy (SQDM), to quantify the surface potentials and extract vertical dipole moments of each structure. These experimental dipole moments are then compared with the charge density and orbital occupations obtained by density functional theory calculations (DFT). The authors attempt to rationalize the evolution of dipole moments by considering three effects being (1) the screening of the metal free electrons leading to the polarization of the surface/adatom electron density, (2) the Smoluchowski effect and (3) substrate-adsorbate charge transfer. When dimers are formed, the charge transfer is impacted by the formation of bonding/antibonding orbitals, that reduces the contributions of (1) and (2) leading to a constant per-atom dipole of Au adatoms in chains of various lengths. Although this mechanism described by the authors is not quantitative, the explanation is plausible to explain the difference of dipole moments between Au and Ag adatoms and their evolution with chain lengths.

To me, the main finding is the observation of opposite dipole polarities for Au and Ag nanostructures on Ag(111), which can only be obtained using the SQDM technique. The DFT calculations nicely supports these experimental observations by explaining how charge transfers modifies the dipole magnitude. The manuscript is well-written and the experimental/theoretical data are accurately described. From a general point of view, this is an interesting application of the SQDM technique to nanoscale science which, to my opinion, shows the potential of the technique and deserves publication. Therefore, I recommend the publication of the present manuscript in Nature Communication after minor revisions.

Authors reply:

We thank the reviewer for their positive assessment of our work. Please find below our response to all points raised by the reviewer and a description of the changes made to the manuscript in order to comply with the reviewer’s recommendations.

Here are few comments:

1- In the introduction, the authors state l 54: “In addition, chains of magnetic adatoms created on the surface of a superconductor are candidates for observing Majorana zero modes [7, 13, 14, 31, 32] the quantum properties of which could be exploited in future devices.”

I am not sure this sentence of the introduction is really relevant for the paper. The authors have neither characterized magnetic atoms nor on a superconducting surface. Besides, the formation of Majorana zero modes in atomic chains is bound to the emergence of Yu-Shiba-Rusinov states, that primarily depends on the magnetic exchange interaction of the atom with the superconductor. For sure, dipole moments and charge transfers of adatoms upon adsorption on the superconductor will play a role to the YSR band diagram. If the authors wish to keep this statement, they should develop their thoughts and be more specific.

Authors reply:

Our intention with the entire paragraph, including the sentence quoted by the reviewer, was to illustrate the broad range of applications of atomic chains in general. However, the case of

magnetic atoms on superconductors is indeed quite far from the specific systems studied in our paper. Therefore, to comply with the reviewer's request, we have decided to remove this sentence from the introduction. Instead, we now briefly discuss quantum corrals as suggested by the reviewer (see below).

Following this comment, I would suggest the authors to discuss the important of dipole moments in atomic structures made atom-by-atom with the aim of confining surface electrons of metals. Quantifying dipole moment per atoms and the underlying mechanism of their formation seems to me more interesting in this context (examples can be found here: <https://www.nature.com/articles/s42254-019-0108-5>).

Authors reply:

We have now followed the reviewer's suggestion and added the following sentence to the introduction

“Furthermore, the surface dipoles of isolated atoms or atomic chains are scattering centres for surface electrons, and the dipole strength could be a way to tune the intensity of the effects observed in quantum corrals and artificial lattices [12].”

2- I149: “Each nanostructure was created in a sufficiently large empty surface region, devoid of any surface or sub-surface defects, such that $V \neq 0$ on the image boundary A (compare Fig. 4), as required for the application of Eq. 3.”

There is a strange typo and the sentence is awkward. The authors should correct the sentence.

Authors reply:

In fact, there is no typo in this sentence. The reviewer is probably referring to $V \neq 0$. We intended this to mean "must be equal to" in a compact notation. To clarify the overall consideration, and to add the information requested by Reviewer #2, we have rewritten and expanded the relevant paragraph. It now reads

“Subsequently, we fabricated chains and clusters by lateral manipulation of individual Ag or Au adatoms with the SPM tip, acquired V^+ and V^- images, and computed the V^* image via Eq. 1. The dipole P_{\perp} is then obtained by integrating over the entire V^* image (Eq. 3). To avoid systematic errors, this image must therefore contain the entire contribution from either the chain or the cluster, but no contribution from any other nanostructure. To achieve this, we assemble the clusters or chains in areas far away from any defects, molecular islands, or step edges, and we record images of sufficient size so that the condition $V^* = 0$ is satisfied at the image boundary. The later conditions requires image sizes between $15 \times 15 \text{ nm}^2$ for adatoms and dimers, and $25 \times 25 \text{ nm}^2$ for the 12-atom chain.”

Reviewer #2

The authors of this manuscript report the results of a combined experimental and theoretical study into the electrostatic potential of atomically well-defined nanostructures on surfaces. In particular, it is shown that scanning quantum dot microscopy (SQDM) can be used to quantitatively measure the dipole moment of atoms, and chains thereof. The experimental findings are reproduced by DFT calculations, and it is claimed that the experimental data can be used as a benchmark for theoretical calculations.

Spatial variations in the charge distributions play a crucial role in various fields, ranging from chemical reactivity/catalysis, to simulating matter with artificial lattices. Consequently, I believe the results presented here are of interest to a broad audience.

The manuscript is clearly written (a general introduction into SQDM is much appreciated, and I believe useful to the non-specialist reader) and the arguments are easy to follow. The claims in the paper are supported by the data presented and the methodology is sound (I have some comments about the theoretical aspects, see below). The methods section, together with the indicated references, provide sufficient detail for others to reproduce the results.

Authors reply:

We thank the reviewer for their positive assessment of our work. Please find below our response to all points raised by the reviewer and a description of the changes made to the manuscript in order to comply with the reviewer's recommendations.

I have a few points which might help improve the manuscript.

- The agreement between experiment and theory is impressive (c.f. Fig.3b). However, to better substantiate the claim that the experimental results enable them to be used as a benchmark for theoretical calculations, I believe it is crucial to explicitly show that the theoretical results are converged with respect to e.g. k-point mesh and basis set for the quantity of interest (dipole moment). Probably, the authors have already performed these checks. It would be good to add those checks to the supplementary information.

Authors reply:

We thank the reviewer for their suggestion. We have indeed checked the convergence and it shows that our calculated values for the dipoles are converged with respect to

- (1) the number of layers in the metal slab (6 layers),
- (2) the number of k-points used to model the unit cell (10 points), and
- (3) the confining cutoff potential for the numerically tabulated basis functions (7 angstroms).

We now provide the corresponding plots in a supplementary information document.

Furthermore, DFT results are typically affected by the (type of) exchange-correction functional. To further substantiate the claim of acting as a benchmark, it would be good to show a case where the converged DFT results do not match the experimental data.

Authors reply:

We agree on the importance of the XC functional for the outcome of a particular DFT calculation, and have therefore chosen the commonly used PBE functional. Nevertheless, we also believe that our data, even without any calculation, would serve as a suitable general benchmark for ab initio methods, e.g., due to its high accuracy and the atomically well-defined underlying configurations. However, in the present manuscript, the DFT calculations do not primarily serve the benchmarking aspect, but their main purpose is to help rationalize and interpret our experimental observations and the inherent trends. Yet, our calculations *do* in fact show the limits of the XC functional used, as they deviate significantly for the per-

atom dipole of long Ag chains (Fig. 3b). Therefore, we do not believe that an explicit counterexample is necessary to establish our data as a benchmark. Finally, there are crucial practical considerations that prevent us from including a full-fledged counterexample in our manuscript. Finding a suitable example would require a systematic extensive search, including convergence checks and computationally expensive structural relaxations for each functional and each geometry. This would

- (1) if included here, strongly shift and blur the focus of the paper, which is on experimental data and their interpretation,
- (2) deserve a separate publication,
- (3) be of interest to a rather specific audience,
- (4) take a very long time and require a lot of computational resources.

While we absolutely agree with the referee on the importance of benchmarking XC functionals on our data, we would like to take a thorough approach, perform a comprehensive investigation and dedicate a future publication to the results. We hope that the referee will understand and appreciate this approach.

- Inhomogeneous charge distributions, as well as the electrostatic force field of individual molecules have been visualized with Kelvin Probe Force Microscopy and Atomic Force Microscopy with charged tips before. It would be beneficial if the authors could include a paragraph describing the merits/limitations of each approach.

Authors reply:

We have included a paragraph dedicated to the comparison asked for by the referee in a previously publication [Wagner et al., *Nat. Mater.* **18**, 853 (2019)]. Since we summarize the SQDM methodology in the present paper as well, we agree that a brief comparison could be made. We have now included the following paragraph at the end of section A.

“In general, SQDM images show greatly enhanced resolution compared to Kelvin Probe Force Microscopy (KPFM) images taken at similar tip-sample separations, because the QD is the only sensitive element and the tip and surface shield the electric fields [19]. At tip-sample separations of a few Angstroms, KPFM and probe-particle measurements with passivated tips provide very high intramolecular resolution [35-37], but the interpretation of these images is complicated as chemical interactions and structural relaxation set in and therefore the measurand does not provide a pure electric potential signal anymore. While SQDM can also resolve intramolecular charge distributions [16], it does so in a non-invasive way at much larger tip-surface separations, precluding the modification of the system to be investigated by the probe tip and preserving quantitative interpretability as a convolution of the surface dipole density (Eq. 2).”

Minor points:

- On page 3, the authors write: 'Each nanostructure was created in a sufficiently large empty surface region, devoid of any surface or sub-surface defects.' What does 'sufficiently large empty surface region' mean concretely?

Authors reply:

Indeed, our manuscript was not specific at this point. Since also Reviewer #1 raised a question about the respective paragraph, we have reformulated and extended our text. The updated version is given in the reply to Reviewer #1 above.

- Some of the affiliations appear identical, e.g. 2 and 9, 3 and 10.

Authors reply:

We thank the reviewer for this hint. This indeed slipped our attention and we have removed the respective duplicates.

Once these points are addressed, I recommend to publish this manuscript.

Reviewer #3

The authors present a combined experimental and theoretical study of the electrostatic potential emanating from single atoms, bottom up built atomic chains and clusters using a scanning probe based method. As correctly argued by the authors, the presence of electrostatic potentials at surfaces plays a crucial role, e.g., in catalysis, but are difficult to determine quantitatively on a nanometer scale. The paper is very well written and intelligible organized. Moreover, it was interesting and enjoyable to read.

Authors reply:

We thank the reviewer for their positive assessment of our work. Please find below our response to all points raised by the reviewer and a description of the changes made to the manuscript in order to comply with the reviewer's recommendations.

In the past, the authors developed and established an ingenious approach to measure the electrostatic potential by attaching a molecule to the tip apex of an STM/AFM tip. Its charge state is then deliberately manipulated to map out the local electrostatic potential. These results can be readily compared to DFT calculation, which allow to obtain the electrostatic surface potential from the total charge density. These results can be used to get an intuitive understanding in terms of electrostatic surface dipoles stemming from polarization effects, Smoluchowski smoothing and charge transfer.

In conclusion, this is an excellent piece of work and thus should be published as it is with only minor changes (see further below). The content of the paper is very useful for experimentalists as well as theoreticians. The latter can actually use such data for benchmarking the DFT calculations to predict or reproduce surface dipole moments, while the former might employ the method to get a better understanding of local variations of the electrostatic potential due to the presence of surface features like atoms, clusters, chains or step edges.

Authors reply:

We thank the reviewer for their enthusiastic assessment and recommendation to accept our work for publication.

Having said this, I would like add some question and comments. They all revolve on the issue of lateral resolution, distance dependence and sensitivity and the comparison between SQDM and KPFM-based LCPD measurements.

1. According to Fig. 9, the dipole moments in clusters or chains should be localized at the individual atoms. However, atomic resolution is not obtained, only a slight modulation of the surface potential map on the 12-atom chain can be seen in Fig. 4c, i.e., maxima at the chain ends and the central part. What limits the lateral resolution of SQDM? Note that I am aware of the data in Ref. [18], where submolecular features are visible on larger molecules, but not with atomic resolution and not with the intramolecular resolution of KPFM-based LCPD measurement shown, e.g., in Nature Nanotechnology 7, 227 (2012).

Authors reply:

Here the reviewer comments on Fig. 9 and raises a question about the lateral resolution of SQDM (also compared to KPFM with passivated tips).

There appears to be a slight overinterpretation of the sketch in Fig. 9, which is a schematic representation of the various contributions to the surface dipole. The arrows in Fig. 9 represent schematic, i.e., not quantitative, projections of the total dipole moment (and its contributions) onto the atoms in the chain. The actual potential along the atomic chain, however, is coupled to the overall charge distribution so that the dipole is actually delocalised along the entire chain (see also the low isosurface charge density difference plots in Fig. 7a,b). In a chain of identical atoms, hence, only the outer atoms can be distinguished because of their change in neighbour count and this is what we see in Fig. 4c.

That said, a slight modulation of the electrostatic potential along the chain would still be expected due to the localized nature of the ion cores. However, this modulation is not comparable to the case of a molecule mentioned by the reviewer, where different atomic species with strongly different electron affinities are chemically bonded. It is not clear whether the much weaker potential modulation along a homogeneous atomic chain could be detected with any scanning probe technique to date.

Since Reviewer #2 asked for a comparison of SQDM to KPFM as well, we now discuss the basic aspects that determine the resolution of SQDM images (QD size and screening) in our response to Reviewer #2 and in the corresponding paragraph added to Section A (see above). The general answer to the reviewer's question regarding lateral resolution is that the resolution of scanning probe microscopy techniques, including SQDM, decreases with increasing tip-surface separation. This is especially true for passivated tips, which allow very small separations and thus can detect short-range chemical interactions. However, it is these interactions that destroy the quantitative interpretability of small-separation, high-resolution KPFM/LCPD images.

Similarly, SQDM can achieve very high resolution at distances where the lower end of the molecular QD begins to interact with the sample through short-range forces. However, similar to high-resolution KPFM, this would lead to structural relaxation of the QD and the sample, and thus compromise the quantitative interpretability of the acquired images. Therefore, we deliberately maintain a distance of about 10 angstroms from the surface to avoid such interactions and obtain images of the undisturbed surface. At these distances, the lateral resolution of SQDM is expected to be better than that of KPFM, allowing, e.g., the quantification of atomic surface dipoles. This can be understood from a combination of Eqs. 2 and 3 of our manuscript.

2. Only laterally resolve images are shown. What about the height dependence of the signal? Are height-dependent data difficult to obtain or hard to evaluate?

Authors reply:

It is not entirely clear what is meant by “height-dependent data”. It is straightforward to record SQDM images at different distances (i.e., tip heights), as shown in [Wagner et al., PRL 115, 026101 (2015)]. This changes the lateral image resolution as mentioned above, although the SQDM image resolution is rather weakly distance-dependent due to the screening of electric fields by the tip and surface [Wagner et al., J. Phys. Cond. Mat. 31, 475901 (2019)]. However, in the end, this aspect is mostly irrelevant since the processing step in which we either integrate or deconvolve the V^* image (Eq. 3 or 2, respectively) recovers genuine surface properties, either the surface dipole or the surface dipole density.

“Height-dependent data” could also refer to the modulation of the QD gating efficiency caused by variations in the sample surface topography (and thus its “height”). The relative gating efficiency is the second of the two secondary measurands (the other being V^*) and can likewise be obtained from the V_+ and V_- images. We do not show such relative gating efficiency images as they do not contain any information relevant to the current paper. However, we will publish all the raw data alongside this manuscript so that it is available to anyone who is interested in its analysis.

3. V^* has been also used to represent the LCPD in high resolution KPFM images on single molecules, as e.g., in the before-mentioned publication Nature Nanotechnology 7, 227 (2012). The difference should be mentioned in the text to avoid confusion, because this quantity has been related to the electrostatic surface potential as well, albeit only qualitatively.

Authors reply:

The reviewer is correct. While V^* does indeed denote the LCPD in the respective KPFM papers, the interpretation of V^* in those papers is conceptually similar to our use in SQDM. In both cases, only variations of V^* in an image are practically considered (in KPFM this is because the absolute values are influenced by the unknown tip shape). Therefore, V^* represents changes in the surface potential *as measured*, i.e. in the imaging plane. Thus, V^* images from both techniques differ only in their lateral resolution and in the uncertainties associated with the interpretation of high resolution KPFM images as discussed above. In response to the reviewer’s request, we have added the following sentence:

“Note that the interpretation of V^* in SQDM and in high-resolution KPFM images [35, 36] is conceptually similar as both describe the measured changes in the surface potential, that is, in the imaging plane.”

Optional issues that could be addressed by the authors and might be insightful for the reader:

4. The authors might want to discuss the differences between their method and LCPD data on molecules with intramolecular resolution (no atomic resolution, but quantitative vs. atomic resolution, but only a qualitative comparison with DFT possible) and why the lateral resolution of the latter method seems to be better (cf. point 1).

Authors reply:

Since this point was likewise raised by Reviewer #2, we have now dedicated a paragraph to this aspect (see response above).

5. On single molecules subatomic atomic charge redistributions were detected using KPFM-based LCPD measurements, e.g., sigma-hole imaging and detection of the quadrupole moment of CO [both discussed in Science 374, 863 (2021)] and pi-hole imaging [Nature Communications 14, 4954 (2023)]. Is it possible to obtain similar sensitivity while keeping the quantitative accuracy with the method presented by the authors?

Authors reply:

The respective images require a very small tip-sample separation, resulting in significant tip-sample forces. These forces would normally lead to structural relaxation of the QD, given the low spring constant of the standing PTCDA [Knol et al., Sci. Adv. 7, eabj9751 (2021)]. However, the experiments mentioned by the reviewer were performed at maximum tip-surface attraction, a condition that would substantially stabilise the QD. Therefore, imaging of the pi-hole with SQDM may be possible. However, it is not clear whether the lack of quantitative accuracy that troubles high-resolution KPFM could be overcome in such SQDM measurements. Another source of uncertainty is the size and anisotropic shape of the QD (here a PTCDA molecule), which is likely to play a role when imaging confined charge distributions at very short QD-sample distances.

REVIEWERS' COMMENTS

Reviewer #1 (Remarks to the Author):

The authors answered my questions in a satisfactory manner and made changes to the manuscript accordingly. I recommend the publication of the manuscript to Nature Communication as is.

Reviewer #2 (Remarks to the Author):

I thank the authors for responding carefully and thoroughly to my comments. I understand - and agree with - the argument not to include a specific counter example of a XC-functional that does not reproduce the experimental findings.

I recommend accepting the manuscript as is.

Reviewer #3 (Remarks to the Author):

All items have been addressed. The added text regarding a clarification of the differences between SQDM and KPFM is very helpful. I strongly recommend publication of the manuscript.